# Prognostic Factors in Patients with Metastatic Spinal Cord Compression Secondary to Lung Cancer—A Retrospective UK Single-Centre Study

**DOI:** 10.3390/cancers15184432

**Published:** 2023-09-06

**Authors:** Anna Vassiliou, Temidayo Osunronbi, Synthia Enyioma, Gerardo Rago, Afroditi Karathanasi, Aruni Ghose, Matin Sheriff, Christos Mikropoulos, Elisabet Sanchez, Michele Moschetta, Cyrus Chargari, Elie Rassy, Stergios Boussios

**Affiliations:** 1Department of Medical Oncology, Medway NHS Foundation Trust, Gillingham ME7 5NY, UKelisabet.sanchez@nhs.net (E.S.); 2Hull University Teaching Hospital NHS Foundation Trust, Hull HU1 3SA, UK; 3Department of Medical Oncology, Barts Cancer Centre, St. Bartholomew’s Hospital, Barts Health NHS Trust, London EC1A 7BE, UK; 4Department of Medical Oncology, Mount Vernon Cancer Centre, East and North Hertfordshire NHS Trust, London HA6 2RN, UK; 5Department of Urology, Medway NHS Foundation Trust, Gillingham ME7 5NY, UK; 6Department of Medical Oncology, St Luke’s Cancer Centre, Royal Surrey Hospital, Guildford GU1 1EB, UK; 7Novartis Institutes for BioMedical Research, CH-4056 Basel, Switzerland; michelemoschetta1@gmail.com; 8Department of Radiation Oncology, Pitié Salpêtrière University Hospital, 75013 Paris, France; cyrus.chargari@aphp.fr; 9Department of Medical Oncology, Gustave Roussy Institut, 94805 Villejuif, France; elie.rassy@hotmail.com; 10Faculty of Life Sciences & Medicine, School of Cancer & Pharmaceutical Sciences, King’s College London, London SE1 9RT, UK; 11Kent Medway Medical School, University of Kent, Canterbury CT2 7LX, UK; 12AELIA Organization, 9th Km Thessaloniki–Thermi, 57001 Thessaloniki, Greece

**Keywords:** metastatic spinal cord compression, lung cancer, survivor, prognostic factors

## Abstract

**Simple Summary:**

Metastatic spinal cord compression (MSCC) is characterised by the compression of the spinal cord due to direct or metastatic spread to the vertebrae, potentially leading to neurological deficits. This condition constitutes an urgent situation in oncology, demanding swift diagnosis and immediate intervention due to the considerable risk of spinal cord damage and irreversible neurological repercussions. Spinal tumours resulting from the metastasis of lung cancer are particularly connected with an unfavourable prognosis, often displaying rapid advancement and limited survival. Treatment approaches encompass a combination of radiotherapy and potential surgery, which are tailored to each patient’s situation. Within this retrospective study, our goal was to pinpoint prognostic elements that impact the survival rates of lung cancer patients experiencing MSCC. Identifying such prognostic factors associated with shorter or longer survival subsequent to MSCC could contribute to tailoring distinct, more or less intensive therapeutic strategies for these individuals.

**Abstract:**

Purpose: Metastatic spinal cord compression (MSCC) is a severe complication of cancer that can lead to irreversible neurological impairment, necessitating prompt recognition and intervention. This retrospective, single-centre study aimed to determine the prognostic factors and survival rates among patients presenting with MSCC secondary to lung cancer. Methods and Materials: We identified 74 patients with epidural metastases-related spinal cord compression and a history of lung cancer through the electronic database of Medway Maritime Hospital in the United Kingdom (UK), spanning the period from April 2016 to September 2021. Among them, 39 were below 55 years old, while 35 were aged 55 years or older; 24 patients were diagnosed with small cell lung cancer (SCLC), and 50 patients had non-small cell lung cancer (NSCLC). Results: The median overall survival (OS) was 5.5 months, with 52 out of 74 patients dying within 6 months of diagnosis with MSCC. For the entire cohort, the statistically significant variables on multi-variate analysis were cancer type (NSCLC had improved OS), the number of involved vertebrae (one to two vertebrae involvement had improved OS), and the time taken to develop motor deficits (≤10 days to develop motor deficits had worsened OS). For the NSCLC cohort, the statistically significant variables on multivariate analysis were molecular alterations (patients with epidermal growth factor receptor (*EGFR*) mutation), pre-treatment ambulatory status, Eastern Cooperative Oncology Group (ECOG) performance status, and the time taken to develop motor deficits. Conclusions: Within the entire cohort, patients diagnosed with NSCLC and spinal metastases affecting one to two vertebrae exhibited enhanced OS. Within the NSCLC subgroup, those with *EGFR* mutations who were ambulatory and possessed an ECOG performance status of 1–2 demonstrated improved OS. In both the entire cohort and the NSCLC subgroup, the development of motor deficits within a period of ≤10 days was associated with poor OS.

## 1. Introduction

Metastatic spinal cord compression (MSCC) is defined as the compression of the spinal cord by direct or metastatic spread to the vertebrae, which could result in neurological deficits [1]. It represents a devastating complication of cancer with an estimated incidence rate of about 5–15% in patients with malignancy [2]. The thoracic vertebrae are the most frequently affected region of the spine, often displaying lesions situated in the posterior half of the vertebral body [3]. This is attributed to the venous drainage route of visceral organs through spinal extradural venous plexuses.

In 10–20% of patients, MSCC represents the primary clinical presentation of malignancy, more commonly in patients with lung, breast and prostate cancer [4]. By contrast, spinal Ewing Sarcoma presents very rarely, and only 69 cases have been reported in the scientific literature [5]. MSCC is an acute oncological emergency necessitating prompt diagnosis and urgent intervention, given the high risk of spinal cord injury and irreversible neurological consequences [6]. Metastatic spinal tumours derived from lung cancer are notably linked to poor prognosis, typically exhibiting rapid progression and short survival [7,8].

The clinical presentation of MSCC ranges from progressive pain and paraesthesia to motor weakness, sphincter dysfunction and paraplegia [9]. The vague and insidious course of the signs of MSCC can lead to delays in diagnosis and the delivery of care [10]. magnetic resonance imaging (MRI) has been found to offer a superior diagnostic evaluation, with a high sensitivity of detection for MSCC in comparison to other imaging modalities. Therefore, it is the recommended diagnostic test in case of suspicious MSCC, preferably with the use of a gadolinium contrast [9,11,12], as also suggested by the National Institute for Health and Care Excellence (NICE) guidelines [13].

Survival rates are poor, with just 10% of patients surviving for 18 months after the initial clinical presentation of MSCC [14,15]. When considering the management plan for symptomatic MSCC, the best results are achieved through a multi-disciplinary approach involving input from oncological, radiological and neurosurgical teams [16]. Different therapeutic goals, such as local tumour control, an improvement in neurological deficits and pain management, require the thorough consideration of individual patient factors like the progression of the tumour to other sites and life expectancy [17]. 

Usually, the treatment aim for MSCC is palliation, focusing on the preservation of neurological function and pain relief. As soon as the diagnosis of MSCC is suspected, a loading dose of high-dose steroids should be offered to all patients presenting with neurological deficits. It seems that the intravenous administration of either 10 mg (low-dose) or 100 mg (high-dose) dexamethasone, both followed by 16 mg of dexamethasone orally per day, is effective and safe for the treatment of MSCC [4]. Following the initiation of steroids, patients are stratified for management with radiotherapy in combination with or without surgery, depending on factors such as pain intensity, neurological status, performance status and treatment history [18]. Research has demonstrated that decompressive laminectomy plus the use of adjuvant radiotherapy is superior to treatment with radiotherapy alone and is advocated in patients who have a greater potential for neurological recovery and a more favourable prognosis [19]. The designed and standard treatment protocol for MSCC in our centre was beneficial in assessing, auditing, and improving the standard of care for the acute management of patients presenting with MSCC [20]. In this retrospective study, we aimed to identify the prognostic factors affecting the survival rates of patients with lung cancer presenting with MSCC. The identification of prognostic factors that are associated with shorter or longer survival after MSCC could help with the development of specific treatment approaches, more or less intensive, for these patients.

## 2. Methods

This is a retrospective study that identified 74 patients with spinal cord compression from epidural metastases via the electronic database of Medway Maritime Hospital, Kent, United Kingdom (UK), from April 2016 to September 2021, which was conducted as part of the evaluation of care quality commission’s five-year strategy, published in April 2022. We retrospectively analysed 12 characteristics for overall survival based on a study conducted by Lei. et al., including age (55 years vs. ≥55 years), gender (female vs. male), smoking status (yes vs. no), lung cancer type (non-small cell vs. small cell), pre-treatment ambulatory status [ambulatory vs. non-ambulatory), Eastern Cooperative Oncology Group (ECOG) performance status (1–2 vs. 3–4), the number of involved vertebrae (1–2 vs. 3–4), pre-treatment chemotherapy (yes vs. no), the interval from suspicion of MSCC to MRI (≤24 h vs. >24 h), the time to developing motor deficits (≤10 days vs. >10 days), the interval from confirmed MSCC to radiotherapy (≤24 h vs. >24 h vs. no radiotherapy), and the interval from confirmed MSCC to surgery (≤6 days vs. >6 days vs. no surgery). We also performed a sub-analysis of the non-small cell lung cancer (NSCLC) cohort presenting with MSCC and retrospectively analysed the characteristics of the histology of NSCLC (adenocarcinoma vs. non-adenocarcinoma) and driver molecular alterations (epidermal growth factor receptor ((*EGFR*)) mutation versus a non-*EGFR* mutation such as anaplastic lymphoma kinase ((ALK)) or kirsten rat sarcoma viral oncogene homolog ((KRAS))). For the present study, we included patients who had a definitive diagnosis of MSCC via an MRI of the spine but excluded any patients who were suspected of having MSCC and not referred to for any imaging. 

### Statistical Analysis

Statistical analysis was performed on IBM^®^ SPSS^®^ Statistics 28 (Windows 10). Overall survival (OS) was defined by the time between diagnosis of MSCC and death and was computed using the Kaplan–Meier method. The primary endpoint was death within 6 months of MSCC diagnosis. Univariate and multivariate Cox regression analyses were performed to investigate the influence of various factors on OS. The prognostic factors that were statistically significant in the univariate analysis were included in the multivariate analysis.

In our analysis of the entire cohort, the baseline factors considered included age, gender, smoking status, lung cancer histological subtype (small cell lung cancer ((SCLC)) versus NSCLC), pre-treatment ambulatory status, ECOG performance status, the number of vertebrae involved, pre-treatment with chemotherapy, the interval from suspicion of MSCC to MRI, the interval from confirmed MSCC to radiotherapy, and the interval from confirmed MSCC to surgery. We repeated our analyses for patients with NSCLC only and considered the following variables in addition to the aforementioned factors: histology (adenocarcinoma versus non-adenocarcinoma), driver molecular alterations (epidermal growth factor receptor ((*EGFR*)) versus non-*EGFR*), and targeted therapy (yes or no).

## 3. Results

### 3.1. Entire Cohort

Seventy-four patients met the inclusion criteria. There were 45 male (60.8%) and 29 female (39.2%) patients. The mean age of the patients was 54.5 ± 2.6 years. In total, 50 patients (67.6%) had NSCLC, while 24 patients (32.4%) had SCLC. Fifty-two patients (70.3%) had one to two vertebrae affected, while 22 patients (29.7%) had three or more vertebrae affected. The median OS was 5.5 months (interquartile range ((IQR)): 4.4–7.8 months). Fifty-two out of the seventy-four patients died within 6 months of MSCC diagnosis (a six-month survival rate of 29.7%). In terms of treatment, radiotherapy was administered to 40 out of 74 patients, while decompressive surgery was performed on 15 out of 74 patients. Furthermore, 34 out of 74 patients did not undergo either radiotherapy or surgery, primarily due to significant comorbidities and/or a compromised performance status. This particular subgroup was referred to the hospital’s palliative care teams as well as community-based care resources.

In the univariate Cox regression analysis, the following factors were associated with prolonged survival time: female (hazard ratio ((HR)) 0.50, 95% confidence interval ((CI)) 0.28–0.90), NSCLC (HR 0.18, 95% CI 0.10–0.32), ambulatory (HR 0.37, 95% CI 0.21–0.65), ECOG performance status 1–2 (HR 0.27, 95% CI 0.16–0.49), one or two vertebrae involved (HR 0.15, 95% CI 0.09–0.28), pre-treatment with chemotherapy (HR 0.24, 95% CI 0.13–0.45), and MRI undertaken within 24 h of suspected MSCC (HR 0.37, 95% CI 0.20–0.66). The patients that received radiotherapy (20 Gy in 5 fractions) within 24 h of confirmed MSCC (HR 0.25, 95% CI 0.10–0.66) were more likely to have a prolonged survival time compared to those that received radiotherapy more than 24 h after the confirmed diagnosis of MSCC. The following factors were associated with shortened survival time: being a smoker (HR 2.83, 95% CI 1.38–5.84]) and developing motor deficits within ten days of the diagnosis of MSCC (HR 7.95, 95% CI 4.20–15.1) (Table 1).

Multivariate Cox regression analysis showed that the following variables independently influenced survival at any given time: lung cancer histological subtype, the number of involved vertebrae and time taken to develop motor deficits. Indeed, patients with NSCLC were 82% less likely to die compared to patients with SCLC (adjusted HR 0.18, 95% CI 0.08–0.39). Those with one or two vertebrae metastatic involvement were 64% less likely to die compared to those with three or more vertebrae involvement (adjusted HR 0.36, 95% CI 0.15–0.87). Finally, the patients who developed a motor deficiency within ten days after MSCC diagnosis were 6.1 times more likely to die compared to those who developed motor deficiencies at 11 or more days post-MSCC diagnosis (adjusted HR 6.06, 95% CI 2.59–14.2) (Table 1). The Kaplan–Meier survival curve for cancer type is shown in Figure 1 below, and the Kaplan–Meier survival curves for the number of vertebrae involved and the time to develop motor deficiencies are shown in Figure A1 and Figure A2, respectively, and found in Appendix A section.

### 3.2. NSCLC Cohort

Fifty patients had NSCLC, and the mean age of the NSCLC cohort was 54.1 ± 2.7 years. There were 22 female (44%) and 28 male (56%) patients. Forty-one patients (82%) had one to two vertebrae affected, while nine patients (18%) had three or more vertebrae affected. Regarding treatment strategies, radiotherapy was employed for 27 out of 50 patients, whereas 15 out of 50 underwent surgical intervention. Moreover, 20 out of 50 patients opted against both radiotherapy and surgery, primarily attributed to substantial comorbidities and/or personal preferences. Their care approach was centred around alleviating symptoms through palliative measures. Twenty patients (20/50; 40%) had EGFR molecular alterations, exhibiting a 90% reduced likelihood of succumbing to MSCC when contrasted with patients possessing KRAS (12/50; 24%) and ALK (2/50; 4%) rearrangements or those with unspecified genetic profiles (16/50; 32%). The median OS in the NSCLC cohort was 5.8 months (IQR: 5.2–8.8 months). Among these 50 patients, 28 died within 6 months of MSCC diagnosis (six-month survival rate 44.0%).

In the univariate Cox regression analysis, the following factors were associated with a prolonged survival time–*EGFR* mutation (HR 0.20, 95% CI 0.07–0.52), ambulatory (HR 0.37, 95% CI 0.18–0.79), ECOG performance status 1–2 (HR 0.23, 95% CI 0.11–0.52), one or two vertebrae involved (HR 0.19, 95% CI 0.08–0.42), pre-treatment with chemotherapy (HR 0.33, 95% CI 0.16–0.70) and targeted therapy (HR 0.27, 95% CI 0.11–0.63). The patients who received radiotherapy within 24 h of confirmed MSCC (HR 0.33, 95% CI 0.12–0.93) and those who did not receive radiotherapy at all (HR 0.42, 95% CI 0.18–0.95) were more likely to have a prolonged survival time compared to those who received radiotherapy more than 24 h after the confirmed diagnosis of MSCC. Here, there are possibly selection biases, as radiotherapy was not proposed to those with potentially limited MSCC. Developing motor deficits within ten days of MSCC diagnosis was associated with a shortened survival time (HR 8.31, 95% CI 3.65–18.9) (Table 2). 

Multivariate Cox regression analysis showed that the following variables independently influenced survival at any given time: the type of molecular alteration, pre-treatment ambulatory status, ECOG performance status, and the time taken to develop motor deficits. Patients with an *EGFR* mutation were 90% less likely to die compared to those with non-*EGFR* mutations (adjusted HR 0.10, 95% CI 0.01–0.70). The patients who had ambulatory pre-treatment were 64% less likely to die compared to those who were non-ambulatory (adjusted HR 0.36, 95% CI 0.14–0.92). Patients with an ECOG performance status of one or two were 71% less likely to die compared to those with an ECOG performance status of three or more (adjusted HR 0.29, 95% CI 0.11–0.80). Those who developed motor deficiencies within ten days of MSCC diagnosis were 6.2 times more likely to die compared to the patients who developed motor deficiencies at 11 or more days post MSCC diagnosis (adjusted HR 6.17, 95% CI 2.14–17.8) (Table 2). The Kaplan–Meier survival curves for molecular alterations, pre-treatment ambulatory status, ECOG performance status, and the time to develop motor deficiencies are shown in Figure A3, Figure A4, Figure A5 and Figure A6, respectively, in section Appendix A section.

## 4. Discussion

MSCC is an oncological emergency for which early detection and prompt management are critical to preserving patients’ neurological function. Lung cancer represents the most common tumour type among patients presenting with MSCC, overall representing approximately 20% of cases [20,21,22,23]. Several studies have shown that the MSCC sequela of lung cancer has worse survival compared to other solid tumours such as prostate and breast cancer [24,25,26,27]. Weigel et al. reported that the survival time of lung cancer patients presenting with MSCC was 2.1 months in comparison to 7.3 months in prostate cancer patients and 21.2 months in breast cancer patients [28]. In this study, we showed that the median survival of lung cancer patients presenting with MSCC was 5.5 months, and the six-month survival rate was 29.7%. 

In the entire lung cancer cohort, we identified two prognostic factors that demonstrated a statistically significant impact on patient survival with NSCLC and one to two affected vertebrae. In the NSCLC subgroup, the following factors were associated with improved overall survival: *EGFR* molecular alterations, ambulatory patients, and the ECOG performance status of 1–2. The early development of motor deficits (≤10 days) was associated with worsened OS in both the entire lung cancer cohort and the NSCLC subgroup. 

There are several scoring systems available to predict survival in patients with spinal metastasis. The Tokuhashi score comprises six factors that are deemed influential for prognosis, encompassing overall health status, the count of bone metastases apart from spinal involvement, the number of spinal metastases, primary lesion type, the presence of major organ metastases, and paralysis status [8]. Survival durations were projected based on the cumulative score utilising prognostic benchmarks. The survival period was up to 6 months for a total score of 0–8, over 6 months for a total score of 9–11, and over 1 year for a total score of 12 or more. The Tokuhashi score did not incorporate any element in relation to treatment interventions and allowed for an adaptable application. Furthermore, the Spine Oncology Study Group (SOSG) developed the Spinal Instability Neoplastic Score (SINS) in 2010 [29]. This scoring system evaluates and assigns scores to 6 variables—lesion location, pain characteristics, bony lesion type, spinal alignment on radiographs, the extent of vertebral body destruction, and involvement of posterolateral spinal elements. The individual scores for each variable were summed to yield a final score ranging from a minimum of 0 to a maximum of 18. A score between 0 and 6 indicates stability, a score between 7 and 12 suggests uncertain (potentially impending) instability and a score between 13 and 18 signifies instability. For scores exceeding 7, it is advisable to seek surgical consultation. However, none of the known scoring systems are particularly tailored to lung cancer patients, despite a few studies focusing on the analysis of prognostic variables in MSCC secondary to lung cancer [30,31]. Nevertheless, many parameters assessed by these scoring systems were also identified in the present study, including the number of vertebrae affected by spinal metastasis.

From a histological perspective, SCLC behaves more aggressively than NSCLC histology, exhibiting early dissemination to distant sites and rapid growth [32]. The distinction between these two types of lung cancer was clearly evident in our study, where, at any given time, SCLC patients were 82% more likely to die compared to NSCLC patients. This aligns with previous studies showing the survival rate of SCLC ranging from 6 days to 20 months [33,34]. Another independent negative prognostic factor noted in our study was the involvement of more than two vertebrae, with patients being 64% more likely to die compared to those with two or fewer vertebra involved. This is also seen in the literature with studies evaluating the number of vertebrae involved as a factor of advanced disease; in univariate analysis, a positive association with survival was noted when one or two vertebrae were involved [21,35].

The treatment of MSCC includes corticosteroids, radiotherapy, and surgery used either alone or in combination [22,35,36,37,38,39]. Irrespective of the palliative nature of this treatment approach, treatment objectives encompass local tumour and pain management, as well as enhancing neurological functions to enhance overall quality of life. However, within this study, we faced limitations when isolating and individually addressing each of these goals. Our study, however, has shown that having decompressive laminectomy, either less or equal to 6 days from confirmed MSCC, confers a significant impact on survival. This is also supported by a recent study by Meyer et al., who found that early surgery within 16 h, or even 12 h in certain cases, could significantly increase the chances for better functional recovery without increasing the risk of complications [40]. Therefore, referring patients with MSCC to surgery as early as possible is critical in the management of these patients. Younsi et al. reported that after decompressive laminectomy, 61% of 101 neurologically impaired MSCC patients were able to walk at discharge [41]. According to their findings, the better the pre-operative ambulatory function of the patient, the higher the survival rates, which is in line with the literature and the data of our study [42,43,44,45].

The molecular screening of NSCLC is vital to guiding the most appropriate therapies, with studies reporting the dysregulation of several receptor tyrosine kinases (RTK) as common tumorigenic mechanisms in NSCLC [46,47,48]. In this study, we investigated for the first time, to the best of our knowledge, the prognostic effect of *EGFR* mutations, including ALK and KRAS rearrangements, in patients with NSCLC and MSCC. We found that patients with *EGFR* mutations (20/50; 40%) were 90% less likely to die from MSCC in comparison to patients with KRAS (12/50, 24%) and ALK (2/50, 4%) rearrangements or with an undetermined genetic status (16/50, 32%). Importantly, we also reported that patients who received targeted molecular therapy for NSCLC had a higher survival rate than patients who did not receive targeted molecular therapy and developed MSCC. Therefore, future research should incorporate the genetic status of patients with MSCC secondary to NSCLC when assessing the prognostic variables affecting survival so that a decision about a patient’s tailored course of treatment can be supported by scientific evidence. 

Our study has several limitations (retrospective analysis, single-centre experience, small sample size and heterogenous patient populations) and larger scale prospective studies must be conducted for the provision of the best evidence-based information for the management of MSCC in patients with lung cancer.

The major role of radiotherapy in MSCC patients is clearly demonstrated either as a post-operative treatment after urgent surgical decompression and stabilisation or upfront for patients who are not eligible or not fit for surgery. Guidelines on the management of complicated bone metastases have been recently published, and the importance of these scoring systems to predict survival after radiotherapy for MSCC has been highlighted [49]. Indeed, the prognosis (at least functional) is different if there is a deformation of the dural sac, contact without the spinal cord, or compression with or without cerebrospinal fluid visualization [50]. This is another limitation of the current retrospective study that such categorization was not available. Furthermore, the elements of the Tokuhashi score and the SINS criteria have not all been considered due to their absence from the Electronic Patient Record (EPR) system. This presents an added constraint within our study, considering the valuable predictive importance of these parameters. Overall, further work is required to better include these scoring systems into treatment algorithms in cohorts of NSCLC patients and validate our findings in an independent cohort.

## 5. Conclusions

MSCC is an oncological emergency with long-term neurological implications requiring urgent diagnosis and treatment. In this study, we identified that for the entire cohort, patients with a diagnosis of NSCLC with one to two vertebrae involvement in spinal metastases showed improved OS. For the NSCLC cohort, we identified that patients with EGFR mutations, ambulatory and with an ECOG performance status of 1–2, had improved OS. For both the entire and NSCLC cohort, developing motor deficits ≤10 days was associated with worse OS. In light of these findings, further large-scale research is required to identify the prognostic factors that are independent predictors of MSCC secondary to lung cancer. This could lead to the creation of quantitative scores incorporating reliable prognostic tools that are essential for the appropriate prediction of functional outcomes. 

## Figures and Tables

**Figure 1 cancers-15-04432-f001:**
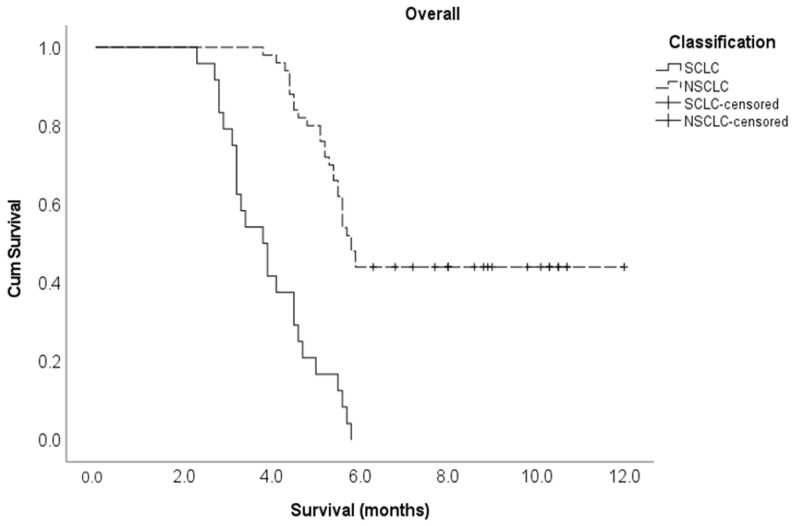
Kaplan–Meier survival curve for cancer type in the entire study cohort.

**Table 1 cancers-15-04432-t001:** Univariate and multivariate Cox regression analysis of baseline characteristics for survival in the entire cohort of lung cancer patients with MSCC.

Characteristics	*n*	Median OS (Months)	Crude Estimates		Adjusted Estimates	
			HR (95% CI)	*p*	HR (95% CI)	*p*
Age						
≤55 years	29	5.6	0.67 (0.39–1.16)	0.153	Not included	
>55 years	45	5.1	1			
Gender						
Female	29	5.8	0.50 (0.28–0.90)	0.021	0.97 (0.46–2.03)	0.934
Male	45	5.1	1		1	
Smoker						
Yes	54	5.1	2.83 (1.38–5.84)	0.005	1.00 (0.39–2.55)	0.997
No	20	7.0	1		1	
Lung cancer subtype						
Non-small cell	50	5.8	0.18 (0.10–0.32)	<0.001	0.18 (0.08–0.39)	<0.001
Small cell	24	3.9	1		1	
Pre-treatment ambulatory status						
Ambulatory	41	5.8	0.37 (0.21–0.65)	<0.001	0.53 (0.26–1.12)	0.096
Not ambulatory	33	4.5	1		1	
ECOG PS						
1–2	50	5.7	0.27 (0.16–0.49)	<0.001	0.78 (0.35–1.77)	0.556
≥3	24	4.2	1		1	
Number of involved vertebrae						
1–2	52	5.8	0.15 (0.09–0.28)	<0.001	0.36 (0.15–0.87)	0.024
≥3	22	3.6	1		1	
Pre-treatment chemotherapy						
Yes	34	7.0	0.24 (0.13–0.45)	<0.001	0.72 (0.33–1.58)	0.411
No	40	4.5	1		1	
Interval from suspicion of MSCC to MRI						
≤24 h	33	5.9	0.37 (0.20–0.66)	<0.001	0.60 (0.29–1.24)	0.165
>24 h	41	4.6	1		1	
Time developing motor deficits						
≤10 days	36	4.4	7.95 (4.20–15.1)	<0.001	6.06 (2.59–14.2)	<0.001
>10 days	38	7.5	1		1	
Interval from confirmed MSCC to radiotherapy						
≤24 h	12	7.4	0.25 (0.10–0.66)	0.009	1.34 (0.40–4.51)	0.529
No radiotherapy	34	5.5	0.56 (0.31–0.99)		0.74 (0.36–1.52)	
>24 h	28	4.8	1		1	
Interval from confirmed MSCC to surgery						
≤6 days	7	8.9	(0.00–)	0.159	Not included	
No surgery	59	5.1	2.72 (0.98–7.57)			
>6 days	8	6.9	1			

Abbreviations: OS, overall survival; *n*, number of patients; HR, hazard ratio; CI, confidence interval; ECOG, Eastern Cooperative Oncology Group; PS, performance status; MRI, magnetic resonance imaging; MSCC, metastatic spinal cord compression.

**Table 2 cancers-15-04432-t002:** Univariate and multivariate Cox regression analysis of baseline characteristics for survival in NSCLC patients with MSCC.

Characteristics	*n*	Median OS (Months)	Crude Estimates		Adjusted Estimates	
			HR (95% CI)	*p*	HR (95% CI)	*p*
Age						
≤55 years	32	5.6	0.76 (0.35–1.62)	0.473	Not included	
>55 years	18	5.1	1			
Gender						
Female	22	5.8	0.48 (0.22–1.06)	0.068	Not included	
Male	28	5.1	1			
Smoker						
Yes	30	5.1	1.68 (0.76–3.73)	0.199	Not included	
No	20	7.0	1			
Histology of NSCLC						
Adenocarcinoma	30	6.5	0.78 (0.37–1.64)	0.511	Not included	
Non-adenocarcinoma	20	5.7	1			
Driver molecular alterations in NSCLC						
*EGFR*	20	8.0	0.20 (0.07–0.52)	0.001	0.10 (0.01–0.70)	0.021
Non-*EGFR*	30	3.9	1		1	
Pre-treatment ambulatory status						
Ambulatory	32	5.8	0.37 (0.18–0.79)	0.010	0.36 (0.14–0.92)	0.032
Not ambulatory	18	4.5	1		1	
ECOG PS						
1–2	38	5.7	0.23 (0.11–0.52)	<0.001	0.29 (0.11–0.80)	0.017
≥3	12	4.2	1		1	
Number of involved vertebrae						
1–2	41	5.8	0.19 (0.08–0.42)	<0.001	1.08 (0.32–3.68)	0.900
≥3	9	3.6	1		1	
Pre-treatment chemotherapy						
Yes	31	7.0	0.33 (0.16–0.70)	0.004	0.49 (0.17–1.39)	0.178
No	19	4.5	1		1	
Interval from suspicion of MSCC to MRI						
≤24 h	29	5.9	0.52 (0.25–1.09)	0.084	Not included	
>24 h	21	4.6	1			
Time developing motor deficits						
≤10 days	18	4.4	8.31 (3.65–18.9)	<0.001	6.17 (2.14–17.8)	<0.001
>10 days	32	7.5	1		1	
Interval from confirmed MSCC to RT						
≤24 h	12	7.4	0.33 (0.12–0.93)	0.043	0.99 (0.29–3.45)	0.902
No radiotherapy	23	5.5	0.42 (0.18–0.95)		0.82 (0.30–2.19)	
>24 h	15	4.8	1		1	
Interval from confirmed MSCC to surgery						
≤6 days	7	8.9	0 (0–)	0.536	Not included	
No surgery	35	5.1	1.83 (0.63–5.29)			
>6 days	8	6.9	1			
Targeted therapy for NSCLC						
Yes	22	8.0	0.27	0.003	2.70 (0.49–15.0)	0.257
No	28	5.4	1		1	

Abbreviations: OS, overall survival; N, number of patients; HR, hazard ratio; CI, confidence interval; NSCLC, non-small cell lung cancer; EGFR: epidermal growth factor receptor; ECOG, Eastern Cooperative Oncology Group; PS, performance status; MRI, magnetic resonance imaging; RT, radiotherapy; MSCC, metastatic spinal cord compression.

## Data Availability

Summary statistics of association analyses are available from the corresponding author upon reasonable request.

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
