# Peer review of "Prognostic Factors in Patients with Metastatic Spinal Cord Compression Secondary to Lung Cancer—A Retrospective UK Single-Centre Study"

_cancers, 2023, doi:10.3390/cancers15184432_

Round 1

Reviewer 1 Report

Review of manuscript “Prognostic factors in patients with metastatic spinal cord compression secondary to lung cancer – a retrospective UK singlecenter study”. The authors present a cohort of 74 patients with epidural lung cancer metastasis causing spinal cord compressions. The authors present interesting findings which can add in the clinical decision making and prognostication of this cohort. I have a few comments.

Introduction

Can you comment why the thoracic spine is most commonly involved? “The area of the spine most commonly found to be involved is the thoracic vertebrae, with lesions usually located in the posterior half of the vertebral body [3].

Results

I can’t find the result of the analysis of EGFR mutations, ALK and KRAS rearrangements? The results are commented on under Discussion but seems to be lacking under Results?

Author Response

Dear Editor and Reviewers,

I am pleased to resubmit for publication the revised version of cancers-2537568 manuscript, entitled “Prognostic factors in patients with metastatic spinal cord compression secondary to lung cancer – a retrospective UK single-center study”.

Thankfully the reviewers provided us with a great deal of guidance, regarding how to better position the article. We are hopeful you agree that this revision will update our comprehensive review. All the comments have been addressed, as shown in the revised version of the manuscript, along with this point-by-point response to the reviewers' comments.

All corresponding are blue changes in the manuscript.

Reviewer #1:

General comment:

Review of manuscript “Prognostic factors in patients with metastatic spinal cord compression secondary to lung cancer – a retrospective UK singlecenter study”. The authors present a cohort of 74 patients with epidural lung cancer metastasis causing spinal cord compressions. The authors present interesting findings which can add in the clinical decision making and prognostication of this cohort. I have a few comments.”.

Response:

Thank you for your positive reinforcement and constructive feedback. We appreciate the opportunity to revise our work for consideration for publication.

Specific comments:

1. Introduction

Can you comment why the thoracic spine is most commonly involved? “The area of the spine most commonly found to be involved is the thoracic vertebrae, with lesions usually located in the posterior half of the vertebral body [3].”

Response:

Thank you for your comment.

We have rephrased as follows, in order to explain why thoracic spine is most often affected.

The area of the spine most commonly found to be involved is the thoracic vertebrae, with lesions usually located in the posterior half of the vertebral body, given the venous drainage of visceral organs through spinal extradural venous plexuses [3].”.

2. Results

I can’t find the result of the analysis of EGFR mutations, ALK and KRAS rearrangements? The results are commented on under Discussion but seems to be lacking under Results?

Response:

Thank you for your recommendation.

We have now incorporated our molecular findings in the “Results – NSCLC cohort” section, as you kindly suggested and rephrased as follows:

Twenty patients (20/50; 40%) had EGFR molecular alterations, exhibiting a 90% reduced likelihood of succumbing to MSCC when contrasted with patients possessing KRAS (12/50; 24%) and ALK (2/50; 4%) rearrangements, or those with unspecified genetic profiles (16/50; 32%). The median OS in the NSCLC cohort was 5.8 months [IQR: 5.2 – 8.8 months]. Among these 50 patients, 28 died within 6 months of MSCC diagnosis (six-month survival rate 44.0%).”.

Reviewer 2 Report

this is a very good, timely and important analysis on a subject with increasing relevance/importance

Author Response

Dear Editor and Reviewers,

I am pleased to resubmit for publication the revised version of cancers-2537568 manuscript, entitled “Prognostic factors in patients with metastatic spinal cord compression secondary to lung cancer – a retrospective UK single-center study”.

Thankfully the reviewers provided us with a great deal of guidance, regarding how to better position the article. We are hopeful you agree that this revision will update our comprehensive review. All the comments have been addressed, as shown in the revised version of the manuscript, along with this point-by-point response to the reviewers' comments.

All corresponding are blue changes in the manuscript.

Reviewer #2:

  • General comment:

this is a very good, timely and important analysis on a subject with increasing relevance/importance”.

Response:

We appreciate you taking the time to review our paper. Thank you for your positive feedback.

Reviewer 3 Report

Interesting topic which merits research projects like the one addressed by the authors, but some described facts should be highlighted:

A hight proportion of patients received no surgical nor radiotherapy treatment:

-          Entire cohort: 34 out of 74 patients did not receive RT and 59 out of 74 patients did not receive surgery

-          NSCLC cohort: 23 out of 50 patients did not receive RT and 35 out of 50 patients did not receive surgery.

The authors should specify the number of patients who did not receive RT nor surgery and explain the reasons for treatment omission. 

Some relevant parameters in MSCC evalutation are not reported like MSCC score and SINS (Spine Instability Neoplastic Score). As those parameters could have prognostic significance, they should be reported. On the opposite, the reasons to omit them should be explained as a study limitation.

Despite palliative goal was the main treatment purpose, functional status, pain control and local tumor control were not reported. That is an important limitation that should prompt some consideration in the discussion.

To find prognostic factors for MSCC patients is the main author's goal, but no mention can be found in the text to already known prognostic score systems (Tokuhashi score, etc). Some items on those prognostic score systems, like number of extraspinal metastatic foci or resectable/unresectable internal organs metastasis are not considered by the authors. To elaborate a more refined prognostic score system including molecular data or treatment quality, as it is the author's intention, should not exclude already known items with prognostic value. Data base should be completed with those items or an explanation for the omission should be included in the discussion.

Author Response

Dear Editor and Reviewers,

I am pleased to resubmit for publication the revised version of cancers-2537568 manuscript, entitled “Prognostic factors in patients with metastatic spinal cord compression secondary to lung cancer – a retrospective UK single-center study”.

Thankfully the reviewers provided us with a great deal of guidance, regarding how to better position the article. We are hopeful you agree that this revision will update our comprehensive review. All the comments have been addressed, as shown in the revised version of the manuscript, along with this point-by-point response to the reviewers' comments.

All corresponding are blue changes in the manuscript.

Reviewer #3:

  • General comment:

Interesting topic which merits research projects like the one addressed by the authors, but some described facts should be highlighted:”

Response:

We appreciate you taking the time to offer us your comments and insights related to the paper. Thank you for your positive reinforcement and constructive feedback. We tried to be responsive to your concerns as we approached our revision.

  • Specific comments:

  1. A hight proportion of patients received no surgical nor radiotherapy treatment:

- Entire cohort: 34 out of 74 patients did not receive RT and 59 out of 74 patients did not receive surgery

- NSCLC cohort: 23 out of 50 patients did not receive RT and 35 out of 50 patients did not receive surgery.

The authors should specify the number of patients who did not receive RT nor surgery and explain the reasons for treatment omission.

Response:

Thank you for your comment. We have now added the relevant information in the “Results” section for the “Entire cohort” and “NSCLC cohort”, (34/74 and 20/50, respectively). Comorbidities, patients’choice and/or frailty were the reasons behind the decision of palliation rather than treatment for this subset of patients.

  1. Some relevant parameters in MSCC evalutation are not reported like MSCC score and SINS (Spine Instability Neoplastic Score). As those parameters could have prognostic significance, they should be reported. On the opposite, the reasons to omit them should be explained as a study limitation.

Response:

Thank you for your valuable comment. We have included a comprehensive depiction of the Spinal Instability Neoplastic Score (SINS) within the third paragraph of the "Discussion" section. Furthermore, in the concluding paragraph of the "Discussion", we have elucidated the rationale behind our omission of these parameters (absence of these data from the Electronic Patient Record system).

  1. Despite palliative goal was the main treatment purpose, functional status, pain control and local tumor control were not reported. That is an important limitation that should prompt some consideration in the discussion.

Response:

Thank you for your consideration. We have made a relevant comment within the fifth paragraph of the "Discussion" section.

  1. To find prognostic factors for MSCC patients is the main author's goal, but no mention can be found in the text to already known prognostic score systems (Tokuhashi score, etc). Some items on those prognostic score systems, like number of extraspinal metastatic foci or resectable/unresectable internal organs metastasis are not considered by the authors. To elaborate a more refined prognostic score system including molecular data or treatment quality, as it is the author's intention, should not exclude already known items with prognostic value. Data base should be completed with those items or an explanation for the omission should be included in the discussion.

Response:

Thank you for your valuable comment. We have now incorporated the Tokuhashi score into the third paragraph of the "Discussion" section. Furthermore, in the concluding paragraph of the "Discussion", we have elucidated the rationale behind our omission of these parameters (absence of these data from the Electronic Patient Record system).

Round 2

Reviewer 3 Report

No aditional comments.